# A Mold Damage Monitoring Algorithm for Power Metallurgy Molding Machines Using Bidirectional Long Short-Term Memory on an Internet of Things Platform [note 1]

**DOI:** 10.3390/s25072143

**Published:** 2025-03-28

**Authors:** Hao-Pu Lin, Yuan-Chieh Chen, Chin-Chuan Han, Yu-Chi Wu, Jin-Yuan Lin

**Affiliations:** 1Ph. D. Program in Material and Chemical Engineering, National United University, MiaoLi 360302, Taiwan; d1012005@o365.nuu.edu.tw; 2Department of Computer Science and Information Engineering, National United University, MiaoLi 360302, Taiwan; tryit320495@gmail.com; 3Department of Electrical Engineering, National United University, MiaoLi 360302, Taiwan; ycwu@nuu.edu.tw (Y.-C.W.); yuan@nuu.edu.tw (J.-Y.L.)

**Keywords:** intelligence system, vibration data, Internet of Things (IoT), deep learning, inertial measurement unit (IMU), mean square error

## Abstract

**Highlights:**

**What are the main findings?**
Damage to the mold will be reflected in the vibration.The vibration caused by the damaged mold is very small.

**What are the implications of the main finding?**
Bidirectional LSTM can be used to determine the mold status through vibration.The accuracy highly depends on the captured data, with a high sampling rate.

**Abstract:**

In this paper, an analysis and monitoring algorithm is proposed for mold health evaluation using vibration data. Two inertial measurement units (IMUs) and an embedded system are first used to acquire vibration data from a powder metallurgy molding machine. These data are collected on an Internet of Things (IoT) platform using the Message Queueing Telemetry Transport (MQTT) protocol. For data analysis, the vibration signal on the *Z* axis is segmented to label the contact section of the upper and middle molds, and the corresponding vibration data of the stamping friction on the X, Y, and Z axes are extracted. Using only historical vibration data from normal stamping, a Bidirectional Long Short-Term Memory (Bi-LSTM) model with an attention mechanism is trained to predict normal stamping vibrations several minutes in advance. By comparing the predicted stamping vibrations with the observed data at the current time, the mean square errors (MSEs) are calculated to evaluate the health status of the mold. Several ablation experiments were conducted to assess the performance of the trained model. The average MSE values for normal samples and abnormal samples were smaller than 0.5 and larger than 1.0, respectively. The experimental results confirm that the trained prediction model and evaluation indicators can effectively notify operators in advance. An early warning system using vibration data for mold damage was successfully implemented, enhancing predictive maintenance.

## 1. Introduction

In response to the demands of Industry 4.0, particularly the shift towards intelligent automation, traditional industries have begun transforming their production processes by incorporating automation, networked systems, data analytics, and intelligent monitoring. This transformation aims to reduce labor dependency, enhance production efficiency, and mitigate operational risks for personnel. In Taiwan, the powder metallurgy industry has been active for more than 40 years. The manufacturing process in this industry involves feeding raw metal powders into a mold, stamping them under a certain pressure using hydraulic machines, forming products of the required shape and size, sintering them at high temperatures, and then performing post-processing, such as reshaping and anti-rust treatment. Traditional mold health monitoring methods rely primarily on manual inspection and periodic maintenance schedules. However, manual inspections are subjective, time-consuming, and highly dependent on operator expertise, making them prone to human error. Additionally, simple threshold-based monitoring systems can only detect severe failures but lack the capability to identify early-stage mold degradation. These methods often result in unexpected mold failures, leading to production downtime and financial losses.

Anomaly detection in vibration data has garnered significant attention due to its critical role in fault diagnosis and structural health monitoring using machine learning and deep learning techniques. While conventional machine learning models, such as support vector machines (SVMs) and decision trees, have been applied for fault diagnosis, they often require handcrafted feature extraction and may struggle with complex, high-dimensional data. For instance, Torres et al. [1] proposed an Isolation Forest algorithm for vibration data analysis to detect anomalies in induction motors within a predictive maintenance context. Adaptive training methods for vibration-based anomaly detection have proven effective in early diagnostics for wind turbine condition monitoring, thereby reducing maintenance costs [2]. Additionally, vibration-based anomaly detection frameworks have been extensively applied to rolling element bearings, successfully identifying distinct fault modes and contributing to the development of robust fault diagnosis systems [3]. In all approaches, the vibration data are first transformed into the frequency domain for feature extraction, and posterior probabilities are calculated for classification using traditional machine learning methods.

Recently, many deep learning-based methods have been proposed for anomaly detection using time-series vibration data across various applications. These methods can be categorized into several network architectures, such as convolutional neural network (CNN) + deep neural network (DNN) [4], Long Short-Term Memory (LSTM) + autoencoder [5,6,7,8], feed-forward NN + SVM [9], LSTM + SVM [10], CNN + LSTM [11,12], bi-directional Long Short-Term Memory (Bi-LSTM) [13,14,15,16,17,18,19], and others. They have been applied to domains such as mechanical and industrial systems, energy and power systems, civil and structural monitoring, aerospace, and more, as shown in Table 1.

Since vibration data are usually acquired in high dimensions, Deac et al. [4] designed a CNN for dimensionality reduction and a DNN for classification in motor bearing condition assessment for fault diagnosis. LSTM networks are well suited for handling time-series data due to their ability to capture and retain long-term dependencies. LSTM-based autoencoders have demonstrated promising results in detecting anomalies in vertical carousel storage systems, particularly by improving data preprocessing to enhance accuracy [5]. Additionally, low-sampling-rate vibration data have been explored using artificial intelligence techniques to address cost and data efficiency challenges, which is particularly valuable for structural fault detection [6]. This LSTM plus autoencoder mechanism is also designed for identifying abnormal patterns in rotary machinery [7] and wind turbines [8], respectively. The integration of turbine data and vibration signals into unified models has further improved the predictive accuracy of wind turbine component failures based on data from the Supervisory Control and Data Acquisition (SCADA) system [9]. In their approaches, a feed-forward neural network (NN) was trained to predict future patterns, and a one-class SVM classifier was subsequently trained for anomaly detection. Similarly, Vos et al. [10] replaced the feed-forward NN module with a deep LSTM network for anomaly detection on datasets of helicopter gearboxes and real helicopter flight tests. Other CNN-LSTM-based network architectures have been proposed by Nguyen-Da et al. [11] and Zhang and Zhou [12]. Hybrid models that combine CNNs and LSTM architectures are proven to be effective in capturing both temporal and spatial features in vibration data for structural health monitoring. Their methods were applied to the monitoring of industrial diesel generators [11] and suspension bridges [12], respectively.

Compared to LSTM, Bi-LSTM captures both forward and backward temporal dependencies. Several Bi-LSTM layers are often stacked to capture more complex patterns and dependencies in sequential data for enhancing model performance. For example, Lee et al. [13] proposed a stacked convolutional Bi-LSTM model for bearing fault diagnosis in rotating machinery. Patra et al. [14] further compared the performance of Bi-LSTM and Gated Recurrent Units (GRU) in anomaly detection for rotating machinery. They showed that Bi-LSTM outperforms GRU in modeling long-term vibration patterns. Similarly, Qiu et al. [15] designed a modified Bi-LSTM network to process rolling bearing vibration data using only long-term memory. In other applications of stacked Bi-LSTM networks, Sharma and Sharma [16] applied it to weather forecasting, while Zhu et al. [17] used it for fault diagnosis of wheelset bearings in high-speed trains.

All used input features, i.e., time-series data and anomaly detection, can be executed in real time. Basically, time-series patterns are processed to extracted the discriminant features between normal or abnormal classes, and the convolutional networks, such as CNN or autoencoder, are further adopted for classification. Most of their approaches are implemented on the Internet of Things (IoT)-driven industrial monitoring. In other applications of anomaly detection using vibration data, vibration analysis is crucial in the aerospace sector for detecting flight anomalies in unmanned aircraft, ensuring operational safety [19]. Vibration data have also been analyzed on datasets of helicopter gearboxes and real helicopter flight tests [10]. Additionally, the temporal convolutional network (TCN) combined with an autoencoder and integrated with cloud computing has enabled real-time anomaly detection in construction vibration monitoring [20]. From the above literature, existing deep learning models for anomaly detection in industrial applications primarily focus on rolling bearings and rotating machinery, with limited research on powder metallurgy mold health monitoring [19].

Molds must be installed by experienced experts before production. During operation, a molding machine performs tens or even hundreds of thousands of stamping cycles for a specific product. However, the timing of mold failure is unpredictable, making it difficult to collect labeled training samples for supervised learning. To tackle this issue, the proposed method compares vibration data collected immediately after mold installation with subsequent vibration data to identify deviations in magnitude. The greater the deviation, the further the mold’s health status deviates from its original condition. Once the deviation surpasses a predefined threshold, an early warning system alerts operators to halt production and inspect the mold to prevent severe damage.

The contributions of the proposed method are summarized as follows:
The IMU-based devices are embedded in an Internet of Things (IoT) platform [21]. This platform ensures scalability and adaptability to different molding machines, allowing manufacturers to apply it across various machine settings with minimal modifications.A vibration-based predictive framework is developed to enable early fault warning before visible damage occurs, thereby reducing unexpected downtime.An unsupervised deep learning approach is adopted for anomaly detection, where the model is trained using only historical normal vibration samples; the labels of samples are unnecessary.The model achieves high anomaly detection accuracy, effectively distinguishing between normal and abnormal conditions based on various metric criteria.A real-time monitoring and early warning system is implemented, integrating IoT-based sensing to provide proactive alerts for preventive maintenance.

The originality of this study is briefly described as follows. As we know, each forming machine is expensive and has been in operation for more than a decade in Taiwanese factories. Most old machines lack digital and network capabilities and do not have any sensors. Currently, mold health monitoring relies on manual inspection of product dimensions, as mentioned above. Modifying these machines by embedding sensors is costly, and manufacturers—primarily small and medium-sized enterprises—are unwilling to discard old machines and purchase new, network-enabled equipment. To address this issue, this study establishes an IoT platform. We designed an IMU-based sensor device that can be externally attached to forming machines to collect vibration data from the molds without modifying any part of the forming machine’s system. This data acquisition device is cost-effective, with two devices installed on each forming machine. A single NodeMCU is responsible for transmitting the vibration data, with a total cost of less than USD 25. The data are sent to the broker on the IoT platform for further analysis. Combined with the subsequent deep learning algorithm, the system can provide early warnings and reduce the likelihood of mold damage. Moreover, the IoT-based environment developed in this study can be scaled to accommodate dozens of forming machines within a factory without requiring complex network configurations. It also supports the flexible addition of new sensors, offering high scalability. Additionally, to the best of our knowledge, no related studies have applied vibration data and deep learning methods to mold damage detection in the powder metallurgy industry. Our research lab is the first to utilize these approaches for mold health monitoring. By combining deep learning, IoT-based sensing, and real-time anomaly detection, this study contributes to the advancement of intelligent predictive maintenance strategies, enhancing manufacturing efficiency and reducing operational risks.

The rest of the paper is organized as follows: The proposed method, including the data acquiring device, signal segmentation, deep network architecture, and the design of warning rules, is described in Section 2. In addition, the ablation experiments conducted to demonstrate the validity of the proposed method are described in Section 3. Finally, the discussions and conclusions are provided in Section 4 and Section 5, respectively.

## 2. Methods

Molds have a finite lifespan, typically lasting for tens or hundreds of thousands of stamping cycles. Once the mold surpasses its operational lifespan or sustains damage, it affects the product’s shape due to the extremely small tolerance range (approximately 0.05 mm to 0.1 mm). Figure 1 illustrates a damaged mold with a horizontal crevice, which is highlighted by a red circle. Mold degradation occurs progressively and is challenging to detect through visual inspection. Furthermore, producing extra spare parts for molds is not feasible due to their high costs. As a result, when a mold is damaged, immediate replacement is often impossible, leading to production delays and significant financial losses. Our observations indicate that subtle variations in vibration patterns occur during the stamping process before mold failure. However, these changes are imperceptible to the human eye, and conventional molding machines lack the capability to detect them or provide early warning signs. To address this challenge, this study proposes a machine learning-based approach to monitor mold health by capturing vibration data generated during stamping. Specifically, three-axis inertial measurement units (IMUs) are installed at the upper and middle mold positions to collect vibration data, which are then used to develop a predictive model for mold health estimation.

The proposed method consists of the following four stages to establish an anomaly detection model trained with vibration data from healthy molds: (1) a vibration data acquisition device, (2) signal segmentation and feature extraction, (3) deep learning-based regression model training, and (4) anomaly monitoring. The flowchart of the proposed method, as shown in Figure 2, is briefly described as follows: First, IMU-based devices are mounted on the upper and middle molds to acquire vibration data from three-axis acceleration at a 500 Hz sampling rate during the stamping process. These data are collected on an IoT platform using the Message Queueing Telemetry Transport (MQTT) protocol. Since the upper mold regularly moves up and down, the corresponding signals on the *Z* axis are segmented to identify the contact section of the two molds based on the detected peaks. Next, feature vectors are extracted from the accelerations of the two IMUs on the X, Y, and Z axes according to the segmented contact section. A Bi-LSTM-based deep model is then trained to construct a regression model that predicts future vibration features using historical data from a healthy mold. The mold status is determined by comparing the predicted features with the currently observed features. If the difference exceeds a predefined threshold, an alarm is triggered. Each stage is described in detail in the following subsections.

### 2.1. Vibration Data Acquisition Device

In this study, an inertial measurement unit (IMU, e.g., MPU6050 [22]) and a NodeMCU comprise vibration data acquisition devices, as shown in Figure 3a. Although MPU6050 is a multi-data sensor, e.g., detecting three-axis acceleration, three-axis rotation speed, and temperature, only the accelerations of the X, Y, and Z axes were acquired in a 500 Hz sampling rate in this study. Peripheral passive components were required to connect the IMU module, as shown on the right-hand side of Figure 3. Pins Vcc, GND, SCL, SDA, and AD0 were used. Here, pins Vcc and GND were connected to a 3.3 V power supply; pins SCL and SDA were used for the I^2^C communication protocol; and pin AD0 identified the I^2^C address that allows multiple sensors to be connected on the same bus. In order to control the accelerometer chip and transmit the collected vibration data through the network, NodeMCU, a Micro Control Unit (MCU) based on the core module ESP-12E, as shown on the left-hand side of Figure 3a, initialized sensor MPU6050, set the sampling rate, and performed the communication in the I^2^C protocol. Due to the communication characteristics, two sensors can be connected by connecting their SDA and SCL together on the same bus topology. The address, either 0x68 or 0x69, identified the specified sensor for data transmission, as shown on the right-hand side of Figure 3b. NodeMCU was responsible for connecting the input/output pins to the sensors, and the power supply was stepped down to supply ESP-12E via AMS1117. For the sake of easy installation, the connection ports were designed in the USB Type-C ports. Type-C cables are easy to obtain and install by non-professionals. In the case of faults or cable disconnections, on-site personnel can quickly troubleshoot and restore the monitoring system. The vibration acquisition devices were installed at the positions of the upper and middle molds on the molding machine, as shown in Figure 4.

The IoT platform [21] operates based on three primary roles: the broker, the publisher, and the subscriber. The publisher sends messages to the broker, attaching specific topic tags. The subscriber, in turn, subscribes to topics of interest from the broker. When the publisher transmits new data associated with a specific topic, the broker forwards it to the relevant subscribers based on their requests. This intermediary approach eliminates the need for devices to directly exchange IP addresses, simplifying network configurations and reducing setup complexity. Additionally, the system is highly scalable, allowing for the seamless addition of new sensors as required, thereby ensuring flexibility for future expansion. This IoT-based data collection platform was constructed as shown in Figure 3c using interconnecting sensors, embedded devices, and servers [23]. It was designed to be a data collection, analysis, and monitoring system for reducing manual monitoring, enabling automated judgment and issuing early warning during the entire operation process. In order to achieve stable data transmission for multiple molding machines and future expandability, the Message Queueing Telemetry Transport (MQTT) protocol [24,25] was adopted for collecting the vibration data from IMUs mounted on the upper and middle molds.

### 2.2. Signal Segmentation and Feature Extraction

The structure of a mold is composed of three parts: the upper, middle, and lower parts. After the mold is filled with powder, a molding machine sequentially performs the following actions of (1) the upper mold moving down, (2) the upper mold/middle mold contacting and stamping, (3) the upper mold moving up, and (4) the extra powder pushing out, as shown in Figure 5. These actions are repeated and are known as a stamping cycle. During the stamping cycle of the molding machine, high pressure is only applied during the contact of the mold, i.e., the upper mold contacts the middle mold. Some signals in a cycle do not contain any discriminant features because they contain the data purely resulting from the movement of the mold. Thus, we must identify the corresponding signals in the stamping operations to analyze the vibration data caused by the increased friction force on the mold. The vibration signal section in which the upper mold contacts with the middle mold is found, the mold is stamped, and the mold is demolded. A graphical illustration is given to show the positions of the upper and middle molds during a stamping cycle (Figure 5). The signals at four positions on the Z axis are acquired and drawn by the rectangles; these are shown in zones 1 to 4 in Figure 6, in which zone 2 is the section used for feature extraction. The stamping cycles are regularly and repeatedly operated throughout the entire production process. The Z axis signals of zones 1 and 3 represent the highest positions of the upper mold. Therefore, the peaks of zones 1 and 3 on the Z axis signals should be detected first. Four steps are executed to detect the peak position in order to identify the contact section on the original Z axis signals. Next, the feature vectors are extracted from the signals on the X, Y, and Z axes according to the segmented contact section, i.e., zone 2.

Signal smoothing: The original signals are smoothed for noise removal using a one-by-three convolutional filter. This operation smooths the signal and effectively modifies the pulses with very large abnormal amplitudes.Stamping segment identification: The mean μ and standard deviation σ of the Z axis vibration data are first calculated. The outliers whose values are out of the range [μ − 4σ, μ + 4σ] are identified. If the outliers are located within an interval, e.g., 4000 sampling points in 8 s, this interval is regarded as the normal stamping process because of the movement of the upper mold. The signals within the interval are subsequently segmented and the peak positions are identified. On the other hand, the molding machine is considered to be in a state of downtime if there are no outliers within the interval.Peak detection: The local maximum value is found to be the vertex in the outlier section in step (b). These are the peaks drawn in the two red circles in Figure 6, which are the highest point of the upper mold during a stamping operation.Feature extraction: Two vibration data acquisition devices are independently mounted at positions on the upper and middle molds. Due to the asynchronous sampling rate of the upper mold’s hydraulic system and sensors, there may be a few milliseconds of difference within the mechanical stamping cycle, and a fixed sampling rate cannot be used. The length between peaks still has a difference of 1–5 data points. The vibration data of the X, Y, and Z axes from two IMUs are extracted to be used as the features for abnormal detection. Since molding machines are repeatedly operated at a regular speed, the contact section—the green box area (zone 2) in Figure 6—is obtained with a fixed offset from the peak position. The local minimal peak in the segment is next identified on the Z axis signal, and 500 points are taken from this point in the forward and backward direction. The corresponding 3000 acceleration vibration values on the X, Y, and Z axes are extracted to be the feature vector, i.e., the green box area in Figure 6 and six feature vectors of two IMUs (mounted on the upper and middle molds), as shown in Figure 7. The vibration data of the stamping friction between the two molds are effectively extracted for the mold’s health evaluation during the upper and middle mold contact process.

### 2.3. Training of Deep Regression Model

Since molds are instantaneously and randomly damaged and the vibration data on mold damage are challenging to collect, supervised machine learning techniques cannot be used to train the two-class classification model, i.e., the normal and abnormal classes. Therefore, only normal data are collected to train a deep regression model using unsupervised learning methods.

We designed a mold prediction model using the training vectors extracted from periodic signals in this study. If a CNN (convolutional neural network) or an attention mechanism is adopted as the training model, too many parameters need to be trained and a significant amount of training time is needed. It is difficult to implement the model in single-board computers or edge computing devices for real-time detection. In addition, there are many types of molding machine molds, and it is impossible to replicate the technology for bearing mechanical life detection [26]. Therefore, we refer to the approach in papers [17,18] and design a network architecture for bearing damage analysis.

In this study, a deep learning regression model was trained to later predict the vibration data. As mentioned in the previous section, the original Z axis data were segmented to identify the segments of feature vectors, and the feature vectors were extracted from the corresponding segmented signal of the X, Y, and Z axes. The input vectors consist of a sequence of temporal patterns. This regression model on temporal patterns is used to analyze sequences of historical patterns for predicting future patterns. Bi-LSTM is adopted to enhance the model’s ability to learn from time-series data. Compared to traditional LSTM, Bi-LSTM captures both forward and backward temporal dependencies, improving the accuracy of anomaly detection. Additionally, a Multi-Head Attention (MHA) mechanism is incorporated into the regression model to strengthen its capacity to capture long-range temporal dependencies. MHA can simultaneously focus on critical features across different time steps, mitigating the vanishing gradient problem and enhancing the model’s ability to recognize anomalous patterns. After training the model, if the prediction closely matches the current observation, the mold is classified as healthy; otherwise, it is classified as abnormal.

In this study, a deep network architecture was used for seismic data regression prediction, as shown in Figure 8 and referred to in [17,18]; the network architecture sequentially comprised one input layer, one Bi-LSTM layer with an attention mechanism, one normalization layer, one feed forward layer, one normalization layer, one Bi-LSTM layer, and one full connected output layer. This network model was a regression model and the scenario for its input and output data is described as follows.

When the feature dimension and time step parameters are set to 6000 and 20, respectively, a data array of 6000 by 20 is input into the “InputLayer”, representing a sequence of extracted vibration data from two IMUs located on the contact sections of the upper and middle molds over the past 20 stamping cycles.The first “Bi-LSTM” layer was configured with 128 units to capture bidirectional dependencies in the time-series through forward and backward computations. This layer outputs a sequence of features for each time step and enhances the representation of temporal dependencies. Additionally, the dropout rate is set to be 0.2 in this module.An attention mechanism was adopted to model global dependencies among high-dimensional features. Four attention heads were utilized to extract long-range interactions between features in the ‘Multi-Head Attention’ layer. Similarly, the dropout rate is set to 0.2 again.Additionally, residual connections from the outputs of the “Multi-Head Attention” and “Bi-LSTM_1” layers, along with layer normalization, are applied to improve training stability and convergence.The “Feed Forward” layer was composed of two fully connected layers with a dimension of 128 and ReLU activation for further enhancing the non-linear representation capacity.Furthermore, residual connections and layer normalization were adopted to improve training stability and convergence.The output of the feed forward layer was processed by the second “Bi-LSTM_2” layer with 64 units. This layer further compressed the feature dimensions and generated the final sequential representation. Here, the dropout rate is set to 0.2.A fully connected layer maps the final representation to a 6000-dimensional output, predicting the corresponding features for the next time step.

This deep learning-based regression model effectively captures long-term dependencies in vibration signals and provides a predictive mechanism for mold health assessment, enabling early detection of potential mold degradation.

### 2.4. Anomaly Monitoring

Considering the current feature vector of vibration data $V_{t}$ at time t and the historical vibration feature vectors $V_{t-s},V_{t-s+2},\ldots ,V_{t-1}$, here, the parameter of time step in the Bi-LSTM model was set to be s. The parameters were input into the designed regression model, and a predicted feature vector ${\hat{V}}_{t}$ was output, as shown in Figure 9. During the training process, a large amount vector sequences were collected for training, and the loss function was defined as the mean square error (MSE) between the predicted vector ${\hat{V}}_{t}$ and the observed feature vectors $V_{t}$. The regression network can be considered as a prediction model of vibration data after p time steps, and we hope the errors between vectors ${\hat{V}}_{t}$ and $V_{t}$ (e.g., MSE $\left({\hat{V}}_{t}-V_{t}\right)$) are as small as possible when the mold is operated in its healthy status. Otherwise, the vibration data $V_{t}$ will be in an abnormal state. To design the warning rule, the input historical feature vectors $V_{t-s},V_{t-s+1},\ldots ,V_{t-1}$ were input into the model, and the predicted output ${\hat{V}}_{t-1}$ was compared with the current vibration data $V_{t}$; when MSE $\left({\hat{V}}_{t-1}-V_{t}\right)<t$ is smaller than a threshold *t*, no alarm was given because of the normal operation, and vice versa. An alarm should be issued due to its abnormal occurrence.

After training the model, it was evaluated using various evaluation metrics, such as MSE, Mean Absolute Error (MAE), and Coefficient of Determination (R-squared). Once the model has been trained and evaluated using the MSE metric criterion, as shown in Figure 10, it can be used for mold health status prediction. During the operation of the stamping process, if the model detects an abnormality in the vibration signal, it can alert the operators to check the mold and prevent mold damage.

## 3. Experimental Results

To assess the effectiveness of the proposed method, several experiments were conducted. First, three toy examples were designed to illustrate the difference between the observed vibration data and the predicted data generated by the deep regression model within a single stamping cycle, as shown in Figure 11, Figure 12 and Figure 13. Here, the configurations of the illustrated examples are set with a time step of 20, an input feature dimension of 6000, and the MSE metric as the criterion. As described in the previous section, historical vibration data from t−20 to t−1 were input into the regression model to predict the current vibration data ${\hat{V}}_{t}$ at time t. Here, feature vectors $V_{t-20},V_{t-19},\ldots ,V_{t-1}$ with a time-step parameter of 20 were extracted from approximately 150 s of past data. The MSE value between the predicted vector ${\hat{V}}_{t}$ and the observed vector $V_{t}$ along the X, Y, and Z axes was computed to evaluate prediction accuracy. Figure 11 presents the evaluation results for the training dataset (i.e., the inside test), where the differences between the observed vibration data and the model’s predictions were compared. The average MSE value for the X, Y, and Z axes was 0.1985, indicating minimal discrepancies. To highlight these differences, the range from dimension 400 to 700 in Figure 11 shows that the variations were not significant. Next, Figure 12 illustrates the evaluation results for vibration data collected during a normal stamping cycle (i.e., the outside test), yielding an MSE value of 0.2551. The observed deviations were also minor, particularly within the time window 400 to 740, as shown in Figure 12. Finally, Figure 13 demonstrates the evaluation results for an abnormal stamping cycle, where the MSE value increased substantially to 1.1076. Notably, significant discrepancies were observed along the X and Y axes, highlighting the model’s sensitivity to abnormal vibration patterns. This MSE value far exceeded those obtained for normal stamping cycles in Figure 11 and Figure 12. These results confirm that the trained model accurately predicts the expected vibration data during normal stamping operations, while effectively distinguishing abnormal conditions.

To further evaluate the model’s performance, a dataset was constructed from stamping process data, including nine hours of vibration data under healthy mold conditions and two hours of data preceding mold replacement. The healthy mold data were divided into the following two mutually exclusive subsets: the first seven hours for training and the remaining two hours for testing. During the training process, 80% of the samples were used for model weight optimization, while 20% were allocated for validation. Following the segmentation and feature extraction steps described in the previous section, a total of 2383 feature vectors were obtained for training, each with a dimensionality of 6000 (comprising 1000 data points per X-Y-Z axis from two IMUs). Additionally, the other two-hour testing dataset containing 720 feature vectors was collected under normal conditions. The collected vibration data typically exhibit a fixed periodic pattern. However, temporary pauses occur when operators temporarily halt the molding machine to inspect and verify whether the stamped products meet quality standards, introducing variations in the number of feature vectors. Furthermore, 547 feature vectors were collected during the two hours preceding mold failure and replacement. This dataset was used to assess the model’s effectiveness in detecting abnormal conditions before mold failure.

Moreover, ablation experiments were conducted to demonstrate the effectiveness and robustness of the proposed method, as shown in Table 2. Table 2 presents the predictive error metrics of the model under different conditions, including mean square error (MSE), Mean Absolute Error (MAE), Coefficient of Determination (R^2^), and Mean Absolute Percentage Error (MAPE). The MSE quantifies the mean of the squared differences between predicted and actual values, where lower values indicate a higher prediction accuracy. The MAE represents the average absolute deviation between predicted and actual values, offering robustness against extreme errors. The R^2^ score in our model reflects the mold’s health status; a higher R^2^ value indicates that the mold is in a healthier condition, with its vibration patterns closely following the expected ones. Conversely, a lower R^2^ value suggests that the mold is deviating from its normal operational state, indicating progressive wear and increased likelihood of failure. Additionally, the MAPE expresses the prediction error as a percentage of actual values, facilitating the comparison of errors across different scales. In addition, two parameters of input patterns, feature dimension and time step, are also varied in the experiments for both healthy and abnormal statuses. The input feature dimensions are set to be 6000, 4800, 3600, 2400, and 1200. This means that 1000, 800, 600, 400, and 200 vibration data points are extracted from each axis of the IMU.

The results indicate that as the feature dimension decreases, the metrics MSE and MAE consistently decline under the “Healthy” condition. For example, when the time step is 20, the MSE decreases from 0.4710 to 0.4463, and the MAE decreases from 0.4113 to 0.4143. Additionally, under the “Abnormal” condition, these errors are significantly higher across all three metrics. This trend suggests that the model effectively distinguishes between normal and abnormal conditions, as greater prediction deviations under the abnormal condition indicate that the model significantly differentiates the observed data from the model’s predictions. Furthermore, R^2^ values are used to evaluate the fitness between the observed data and the model’s predictions, where higher values indicate a better model fit. The results show notably lower R^2^ values under the “Abnormal” condition, ranging from approximately −4.4492 to −7.0028. In contrast, the “Healthy” condition exhibits significantly higher R^2^ values, ranging from −0.9682 to −0.7781. This indicates that the model effectively differentiates between normal and abnormal conditions rather than merely fitting all data points. However, in this study, MAPE does not demonstrate strong discriminative ability between healthy and abnormal states compared to MSE or MAE. The MAPE metric is highly sensitive to extreme values and small actual values. It is also undefined for zero actual values and becomes problematic when actual values are close to zero. Based on the observations in Figure 11, Figure 12 and Figure 13 and the statistical data of actual values, more than 75% of the actuals are zero or near zero.

On the other hand, the choice of feature dimension may influence prediction accuracy. Higher-dimensional data may contain information beyond the critical section of compression, potentially affecting the model’s performance. However, the results are not conclusive. In summary, the errors before replacing the damaged mold are much higher than those observed under normal stamping operations across various parameters.

To clearly visualize the sequential evolution of MSE values, we graphically represent the MSE values of the healthy mold (Figure 14a) and the damaged mold (Figure 14b) over time. We observed that most MSE values in Figure 14a are below 0.75, with occasional slight deviations exceeding 1.0. However, in Figure 14b, there is one point where the MSE value exceeds 12 (indicated by a red circle), which is suspected to correspond to the moment when the mold was damaged. Additionally, the frequency of MSE values exceeding 0.75 in Figure 14b is much higher than in Figure 14a. From Figure 14b, there is a significant change in MSE values before the mold is damaged, indicating a gradual decline in the prediction accuracy. To design warning rules with a more effective performance and fewer false alarms, we applied a minimal filter of size 1 by 3, selecting the minimal MSE values within three cycles in both Figure 14a,b. The filtered results are displayed in Figure 14c,d. While some sporadic false alarms are present, their frequency in Figure 14c is lower than that in Figure 14d. Based on Figure 14c,d, the warning conditions are designed to trigger an alert and notify operators when the filtered MSE value exceeds 0.75.

The deep learning architecture, inspired by [17,18], is designed to train a prediction model for the current observed pattern based on historical data from normal stamping cycles. According to the results in [17,18], Bi-LSTM extracts more discriminative features from input patterns over a longer range in both directions. As we know, there are also many deep learning models available for time-series pattern prediction. To evaluate their effectiveness, we modified the network in Figure 8 by replacing the Bi-LSTM module with two widely used models: LSTM [27] and GRU [28]. The unit sizes in all three modules are set as similarly as possible to ensure a fair comparison. Additionally, all parameters, including feature dimensionality, time step, and distance metrics, are kept identical to those in Table 2. The results for the LSTM and GRU modules are presented in Table 3 and Table 4, respectively. Here, the MAPE metric is excluded due to its limited discriminating ability, as discussed previously. Overall, all three modules effectively evaluate the healthy status of the mold, as shown in Table 2, Table 3 and Table 4. To further analyze their performance, we design a simple separation index. As we know, the error values between predictions and observations should be as small as possible for a healthy mold and as large as possible for an abnormal state. The greater the difference in errors between the healthy and abnormal states, the better the model’s distinguishing capability. For example, the errors of Bi-LSTM, LSTM, and GRU under healthy and abnormal conditions are (0.4710, 1.3161), (0.4669, 1.2281), and (0.4728, 1.2703), respectively, when using the MSE metric, with a feature dimension of 6000 and a time step of 20. The corresponding separation degrees are 0.8451, 0.7612, and 0.7975, respectively, by calculating the distance under the healthy and abnormal conditions. These results indicate that the Bi-LSTM module achieves better separation than the other two models. The separation index is set to one if the Bi-LSTM module performs better; otherwise, it is set to zero. All separation indexes are computed to compare the Bi-LSTM module with LSTM and GRU, as shown in Table 5. From this table, we can see that Bi-LSTM is more effective than LSTM and GRU, particularly when using the R^2^ metric.

On the other hand, the execution time for each prediction is analyzed as follows: First, the workspace for the prediction model is set up on a general personal computer equipped with an Intel Core i9-11900 processor, 64 GB of RAM, and Windows 11 operating system. Due to the initialization overhead of the GPU workspace, the prediction time for the first sample exceeds 1 s. After that, the execution times of each prediction for modules Bi-LSTM, LSTM and GRU are reduced to 0.0866, 0.0892, and 0.0873 s, with the standard deviation 0.0345, 0.0541, and 0.0477, respectively.

## 4. Discussion

In this study, we have identified the critical vibration patterns that are associated with mold damage in powder metallurgy molding machines. One notable observation was the presence of a small secondary wave pattern in the Z axis acceleration during the downward motion of a damaged upper mold. This anomaly indicates the section of mold contact. By carefully analyzing the experimental results in Table 2, several key findings of this research are as follows:Effective Early Fault Detection: By acquiring vibration data along the X, Y, and Z axes using IMU sensors and estimating a prediction function through a deep learning model, the system successfully detects early signs of mold deterioration (see Figure 14). The results indicate that abnormal stamping cycles exhibit significantly higher error values compared to normal cycles across various metrics in Table 2, demonstrating the model’s ability to differentiate between healthy and damaged molds.High Prediction Accuracy and Fewer False Alarm: The system’s MSE value for normal stamping data remains as low as 0.6442, while for abnormal conditions, the MSE increases to 1.1344, further validating the effectiveness of the predictive approach. Additionally, the MAE value for normal samples is 0.3548, whereas for damaged molds, it rises to 0.7406, confirming that the system effectively detects anomalies.Robustness of Trained Model: From the ablation experiments, the trained model demonstrated a strong robustness to various parameters, such as feature dimension, metric criteria, and time step. These parameters had minimal impact on prediction accuracy, indicating the model’s robust design.Robust and Scalable Monitoring: The model was trained and evaluated over a long period, using 2383 feature vectors from healthy mold operations and 547 feature vectors collected from the two hours preceding mold failure, demonstrating its robustness in identifying deviations before mold breakdown. Furthermore, the system adapts to different molding machines and can be extended by adjusting feature extraction methods and training datasets.Real-time Monitoring Feasibility: The model’s inference time per sample needs 0.0866 s after initialization and optimization, demonstrating its capability for real-time deployment in industries.

## 5. Conclusions

This study developed a deep learning-based mold health monitoring system using vibration data to detect mold damage in powder metallurgy molding machines. An IMU plus NodeMCU module was designed for data acquisition, capturing vibration signals along the X, Y, and Z axes on an IoT platform. For data analysis, *Z* axis vibration data were used to mark the contact sections of the upper and middle molds, while friction vibration signals on the X, Y, and Z axes were captured during the stamping process. A Bi-LSTM regression model with an attention mechanism was trained using only historical normal vibration data to predict future vibration patterns. From the experimental results, the error values for normal and abnormal samples are as follows: less than 0.5 and greater than 1.0 on the MSE metric, less than 0.45 and greater than 0.70 on the MAE metric, and greater than −1.0 and less than −4.0 on the R^2^ metric, respectively. Observing Table 2, we find that the errors are small under healthy conditions, whereas the errors are large under abnormal conditions and are clearly separated. The threshold value for issuing an alarm, such as the middle value, can be easily determined. This early warning system effectively reduces the risk of mold damage and decreases machine downtime.

This research establishes a foundation for intelligent predictive maintenance in industrial manufacturing, enhancing efficiency and reducing production losses. Despite these promising findings, this study highlights several areas for future exploration. Firstly, while the current dataset effectively demonstrated the feasibility of mold health monitoring, expanding the dataset to include more examples of mold damage under varying operational conditions is crucial. This expansion would enhance the model’s generalizability and its ability to handle the diverse scenarios encountered in real-world production. Second, we believe that increasing the sampling rate from 500 Hz to 1 KHz would be highly beneficial for segmenting the *Z* axis signal. This enhancement would allow for the more precise detection of the contact section between the upper and middle molds. The corresponding extracted features from this section could be used to train the prediction model more effectively, further improving the accuracy of anomaly detection. Next, edge computing could be explored for real-time on-site processing, reducing reliance on centralized computing. Finally, the development of an adaptive early warning system, fine-tuned to specific mold types and operational environments, would be a valuable extension of this research. Such systems could optimize maintenance schedules, minimize downtime, and further validate the practical application of this technology in industrial applications.

## 6. Patents

Based on this work, a patent application (# 112132226) has been filed to the Intellectual Property Office, Ministry of Economic Affairs, Taiwan, R.O.C.

## Figures and Tables

**Figure 1 sensors-25-02143-f001:**
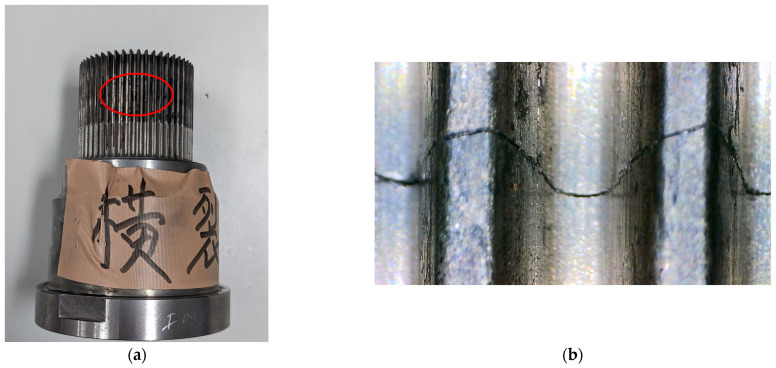
The damaged mold. (**a**) A horizontal crevice (Chinese words) can be seen in the red circle; (**b**) A close-up image of the crevice.

**Figure 2 sensors-25-02143-f002:**
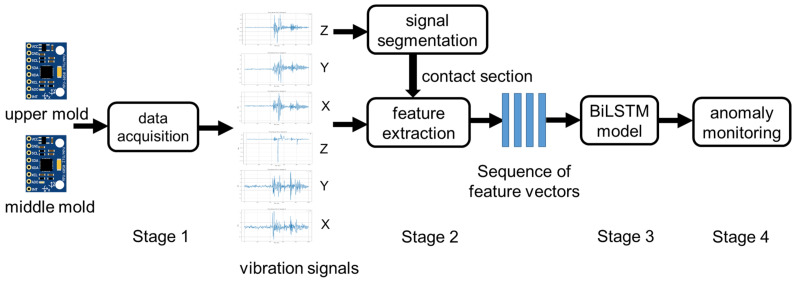
A diagram of the proposed method.

**Figure 3 sensors-25-02143-f003:**
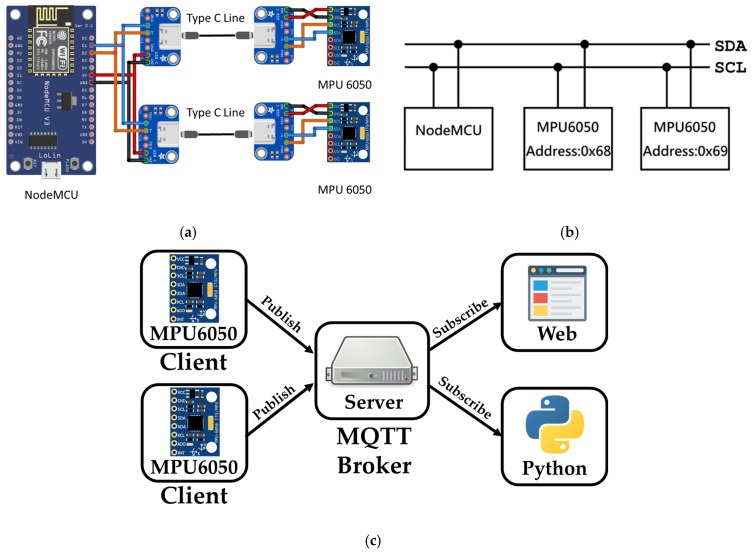
The vibration data acquisition devices and data collection platform. (**a**) Data acquisition hardware; (**b**) Data bus; (**c**) Platform based on the MQTT protocol.

**Figure 4 sensors-25-02143-f004:**
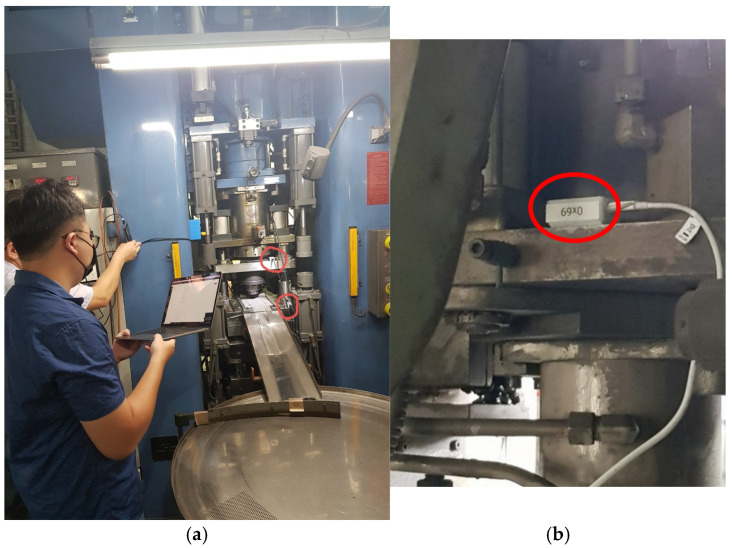
The mounted vibration data acquisition devices; red circles depict the positions of the upper and middle molds. (**a**) Front view of the molding machine; (**b**) Close-up shot of the upper mold.

**Figure 5 sensors-25-02143-f005:**
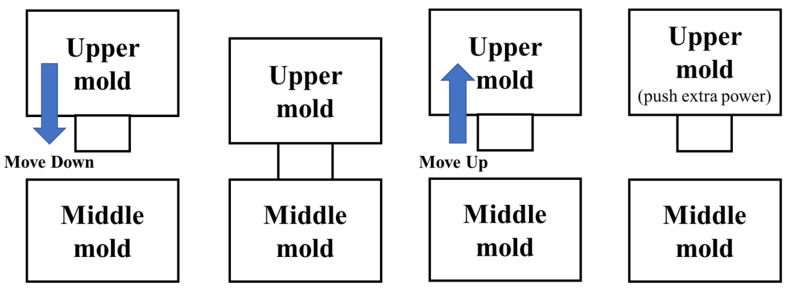
The stamping cycle of the upper mold.

**Figure 6 sensors-25-02143-f006:**
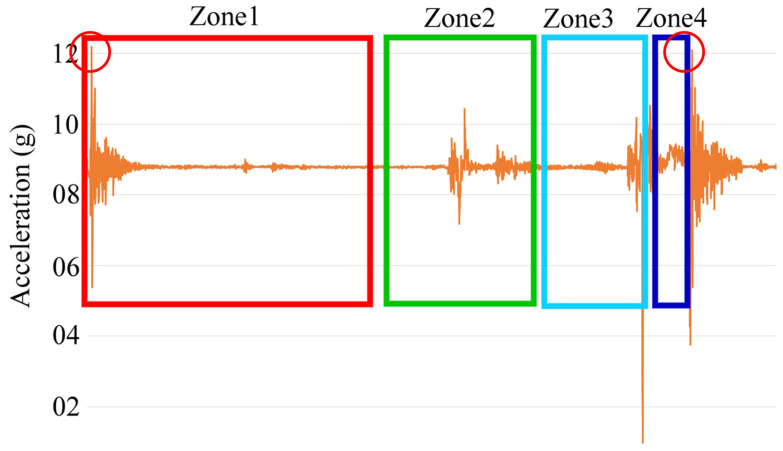
The Z axis vibration signal in a stamping cycle. Two peaks are detected in Step 3 as drawn in the two red circles.

**Figure 7 sensors-25-02143-f007:**
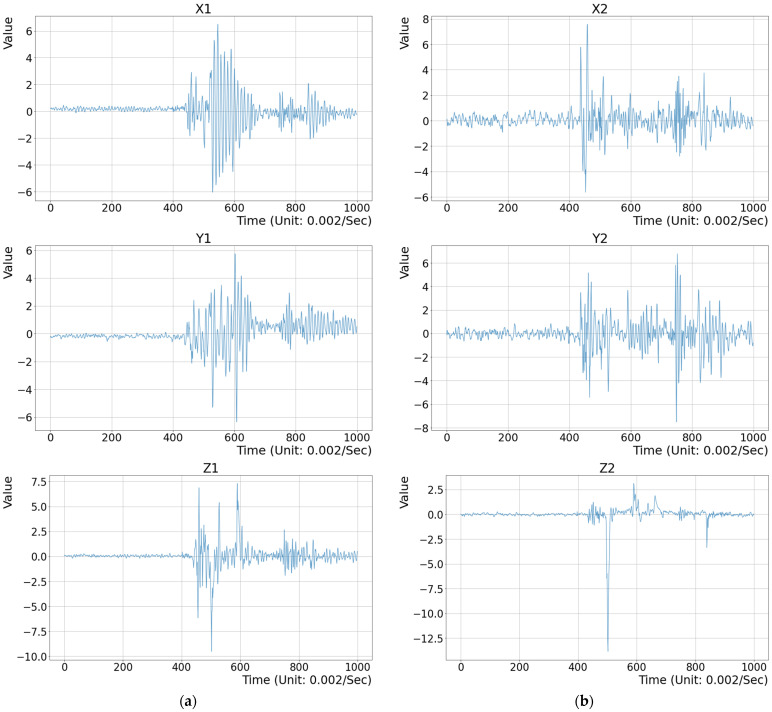
The extracted vibration signals on the X, Y, and Z axes for feature extraction. (**a**) The signals of the IMU on the upper mold; (**b**) The signals of the IMU on the middle mold.

**Figure 8 sensors-25-02143-f008:**
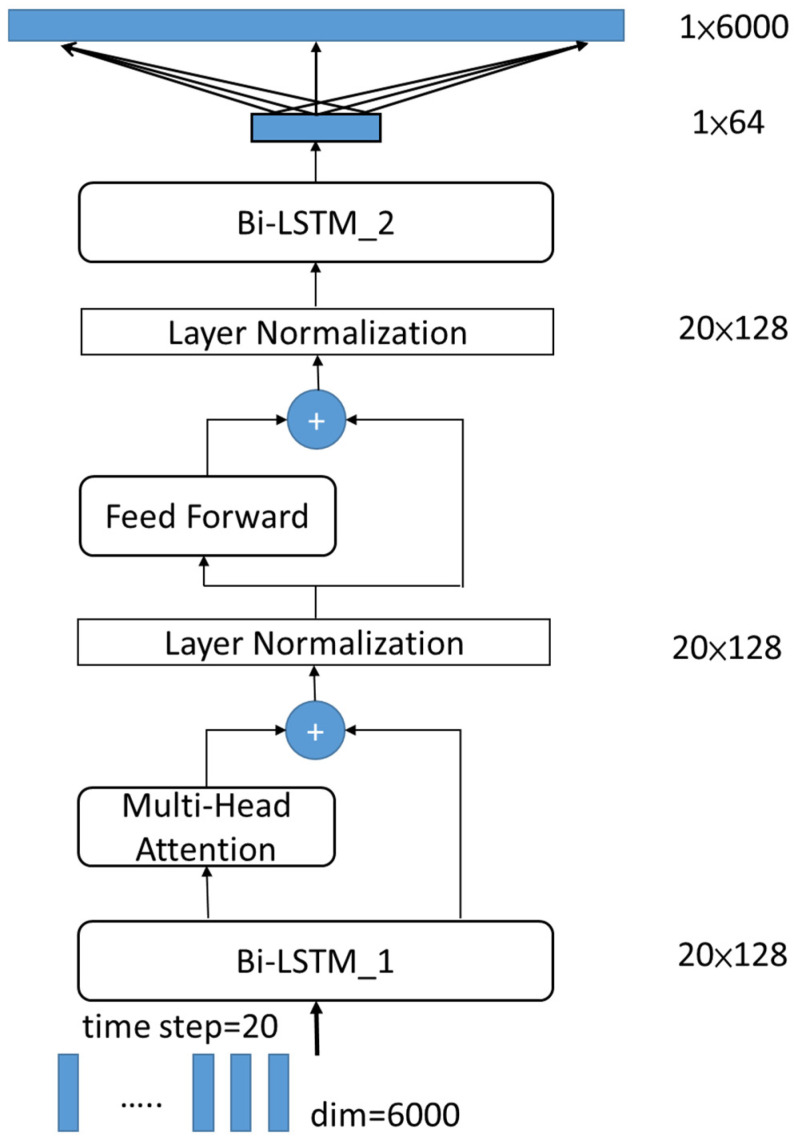
The network architecture of Bi-LSTM with an attention mechanism when parameters of ‘feature dimension’ and ‘time step’ are 6000 and 20, respectively. Here, symbol ‘+’ represents an addition operation.

**Figure 9 sensors-25-02143-f009:**
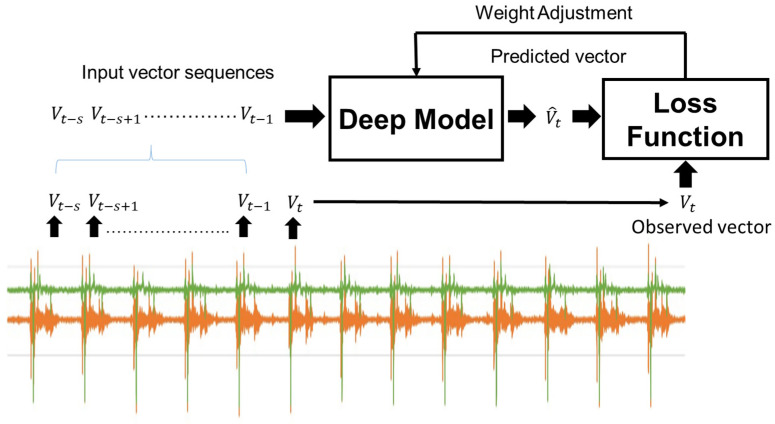
The training process of the deep model.

**Figure 10 sensors-25-02143-f010:**
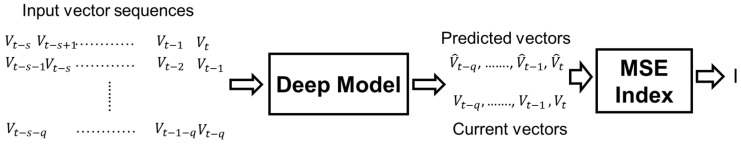
The prediction process of the deep model.

**Figure 11 sensors-25-02143-f011:**
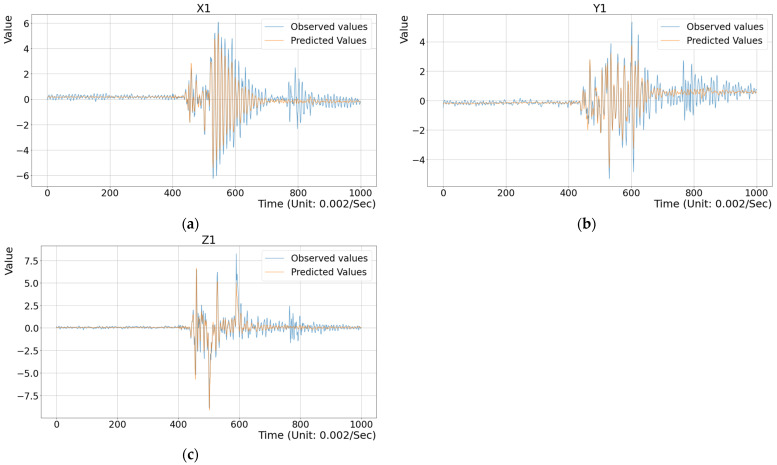
A comparison between the predicted and observed data with an MSE value = 0.1985 in the training dataset (inside test). (**a**) The data on the X axis; (**b**) The data on the Y axis; (**c**) The data on the Z axis.

**Figure 12 sensors-25-02143-f012:**
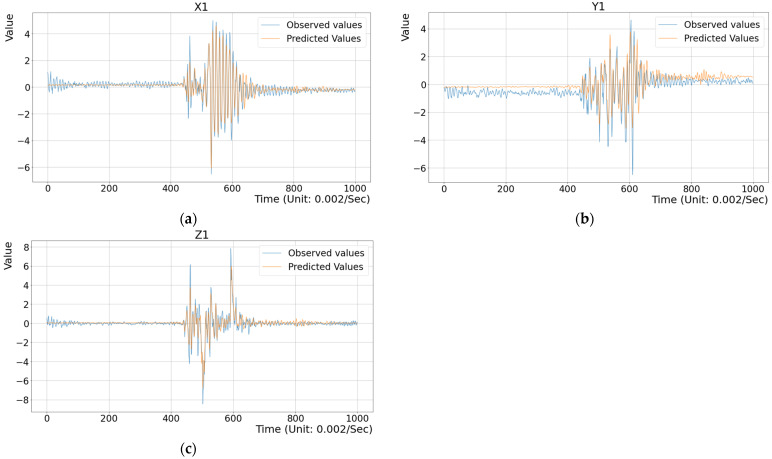
A comparison between the predicted and the observed data under the healthy condition with an MSE value = 0.2551 (outside test). (**a**) The data on the X axis; (**b**) The data on the Y axis; (**c**) The data on the Z axis.

**Figure 13 sensors-25-02143-f013:**
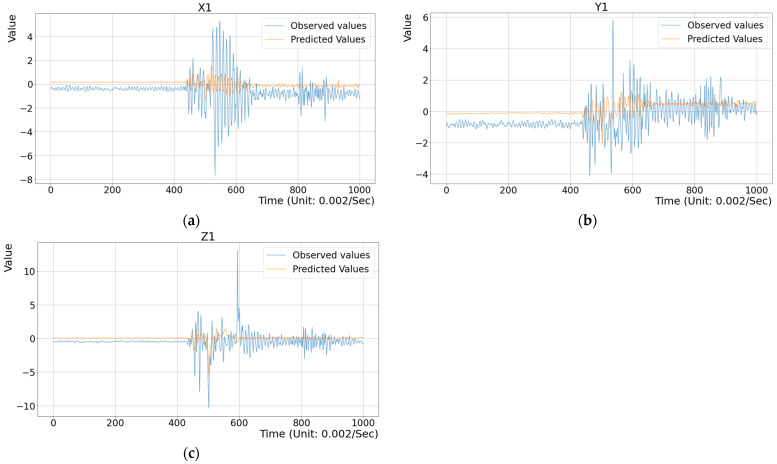
A comparison between the predicted and observed data in the damaged condition with an MSE value = 1.1076. (**a**) The data on the X axis; (**b**) The data on the Y axis; (**c**) The data on the Z axis.

**Figure 14 sensors-25-02143-f014:**
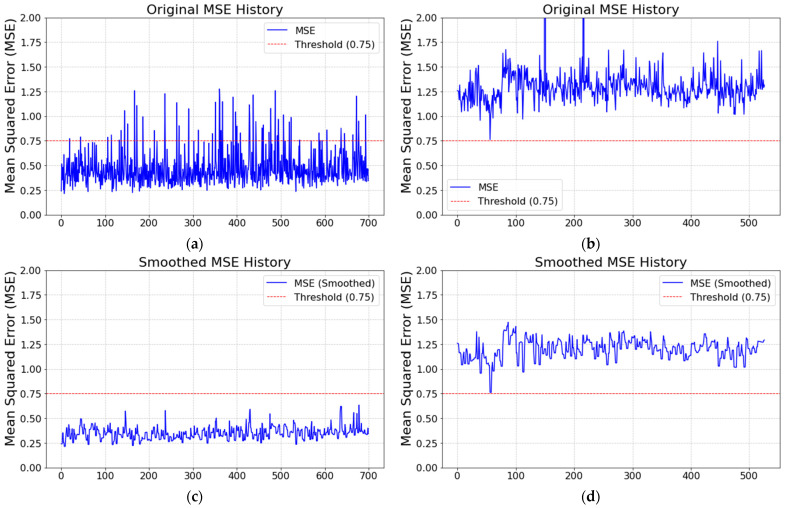
The MSE values between the predicted vectors and observed vectors. (**a**) The MSE values in the health status over two hours; (**b**) The MSE values in the damage status over two hours; (**c**) The MSE values of (**a**) after filtering with a 1 × 3 window size; (**d**) The MSE values of (**b**) after filtering with a 1 × 3 window size.

**Table 1 sensors-25-02143-t001:** Anomaly detection using deep learning methods and vibration data across various applications.

Network Architectures	Applications
CNN + DNN	motor bearing [4]
LSTM + Autoencoder	carousel storage system [5], motor bearing [6], rotary machinery [7], wind turbine [8]
feed-forward NN + SVM	generator bearing of wind turbine [9]
LSTM + SVM	helicopter gearboxes and real helicopter flight tests [10]
CNN + LSTM	diesel generator [11], suspension bridge [12]
Bi-directional LSTM (Bi-LSTM)	bearing of rotating machinery [13], rotating machinery [14], rolling bearing [15], weather forecast [16], wheelset rearing in high-speed trains [17], unmanned aircraft [18], mold damage monitoring [19]
TCN + autoencoder	construction vibration monitoring of buildings [20]

**Table 2 sensors-25-02143-t002:** The results of ablation experiments using the Bi-LSTM module.

	Status, Time Step	Healthy			Abnormal		
Metric, Dimension		10	20	30	10	20	30
MSE	6000	0.4924	0.4710	0.4777	1.2870	1.3161	1.2443
	4800	0.4687	0.4686	0.4658	1.2343	1.2078	1.3077
	3600	0.4532	0.4382	0.4286	1.2381	1.2639	1.2280
	2400	0.4158	0.3988	0.4355	1.2918	1.2390	1.2386
	1200	0.4445	0.4463	0.4303	1.2322	1.1716	1.2633
MAE	6000	0.4175	0.4113	0.4137	0.7406	0.7472	0.7309
	4800	0.4080	0.4081	0.4068	0.7258	0.7207	0.7409
	3600	0.3916	0.3886	0.3863	0.7116	0.7238	0.7125
	2400	0.3844	0.3801	0.3983	0.7229	0.7124	0.7089
	1200	0.4154	0.4143	0.4063	0.7437	0.7209	0.7564
R^2^	6000	−0.9626	−0.9682	−0.9740	−6.7291	−7.0028	−7.2110
	4800	−0.9085	−0.9489	−0.9607	−6.7984	−7.0538	−7.2994
	3600	−0.9259	−0.9449	−0.9279	−6.6974	−6.9014	−7.0825
	2400	−0.9469	−0.8846	−0.9521	−6.1434	−6.2602	−6.4312
	1200	−0.8180	−0.7781	−0.7324	−4.4574	−4.4492	−4.9106
MAPE (%)	6000	248.24	255.55	256.44	252.56	257.75	228.05
	4800	254.00	266.60	267.79	251.29	246.61	262.76
	3600	384.02	379.21	372.47	398.72	438.60	414.73
	2400	409.71	371.38	360.61	371.88	335.70	316.06
	1200	343.15	343.18	317.13	465.26	435.89	482.07

**Table 3 sensors-25-02143-t003:** The results of ablation experiments using the LSTM module.

	Status, Time Step	Healthy			Abnormal		
Metric, Dimension		10	20	30	10	20	30
MSE	6000	0.4677	0.4669	0.4646	1.2393	1.2281	1.2859
	4800	0.4592	0.4529	0.4629	1.2193	1.1516	1.2238
	3600	0.4231	0.4165	0.4173	1.2091	1.1842	1.2120
	2400	0.3803	0.3798	0.3738	1.2487	1.2325	1.2420
	1200	0.3698	0.3698	0.3703	1.2020	1.2020	1.1951
MAE	6000	0.4103	0.4099	0.4105	0.7278	0.7279	0.7423
	4800	0.4046	0.4042	0.4085	0.7233	0.7060	0.7218
	3600	0.3872	0.3801	0.3839	0.7050	0.7015	0.7102
	2400	0.3726	0.3738	0.3724	0.7100	0.7064	0.7142
	1200	0.3813	0.3813	0.3806	0.7286	0.7286	0.7276
R^2^	6000	−0.9274	−0.9452	−0.9528	−6.6675	−6.8689	−7.1395
	4800	−0.9266	−0.9294	−0.9607	−6.8421	−6.9253	−7.2647
	3600	−0.9001	−0.8985	−0.9244	−6.5607	−6.8211	−7.0496
	2400	−0.8593	−0.8719	−0.8566	−6.0252	−6.2414	−6.5422
	1200	−0.6363	−0.6363	−0.6349	−4.3503	−4.3503	−4.6468

**Table 4 sensors-25-02143-t004:** The results of ablation experiments using the GRU module.

	Status, Time Step	Healthy			Abnormal		
Metric, Dimension		10	20	30	10	20	30
MSE	6000	0.4667	0.4728	0.4641	1.2971	1.2703	1.2699
	4800	0.4639	0.4572	0.4601	1.2290	1.1947	1.2060
	3600	0.4370	0.4425	0.4217	1.2378	1.2179	1.2088
	2400	0.3898	0.4069	0.4026	1.2254	1.2071	1.2274
	1200	0.4113	0.4242	0.4163	1.2761	1.2056	1.1605
MAE	6000	0.4108	0.4120	0.4110	0.7421	0.7367	0.7374
	4800	0.4060	0.4054	0.4060	0.7222	0.7160	0.7212
	3600	0.3886	0.3937	0.3850	0.7138	0.7112	0.7121
	2400	0.3754	0.3854	0.3835	0.7086	0.7019	0.7113
	1200	0.4025	0.4100	0.4052	0.7549	0.7294	0.7141
R^2^	6000	−0.9649	−0.9621	−0.9724	−6.7546	−6.9372	−7.1268
	4800	−0.9418	−0.9613	−0.9676	−6.7956	−6.9987	−7.2716
	3600	−0.9436	−0.9591	−0.9344	−6.6405	−6.9001	−7.1006
	2400	−0.8935	−0.9235	−0.9108	−6.0634	−6.2334	−6.4430
	1200	−0.7150	−0.7236	−0.7117	−4.4663	−4.3689	−4.5842

**Table 5 sensors-25-02143-t005:** The separation indexes of Bi-LSTM module compared with modules LSTM and GRU.

	Status, Time Step	LSTM			GRU		
Metric, Dimension		10	20	30	10	20	30
MSE	6000	1	1	0	0	1	0
	4800	1	1	1	1	1	1
	3600	0	1	1	0	1	1
	2400	1	0	0	1	1	0
	1200	0	0	1	0	0	1
MAE	6000	1	1	0	0	1	0
	4800	0	1	1	1	1	1
	3600	1	1	0	0	1	0
	2400	1	0	0	1	1	0
	1200	0	0	1	0	0	1
R^2^	6000	1	1	1	0	1	1
	4800	0	1	1	1	1	1
	3600	1	1	1	1	1	0
	2400	1	1	0	1	1	0
	1200	0	0	1	0	1	1

## Data Availability

The datasets presented in this article are not readily available because the data are part of an ongoing study.

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
