# Peer review of "A Mold Damage Monitoring Algorithm for Power Metallurgy Molding Machines Using Bidirectional Long Short-Term Memory on an Internet of Things Platformâ€"

_sensors, 2025, doi:10.3390/s25072143_

Round 1
Reviewer 1 Report
Comments and Suggestions for Authors
For mold 20 health evaluation using the vibration data, an analysis and monitoring algorithm is proposed, which uses two inertial measurement units and two inertial measurement units to collect vibration data. The effectiveness of the proposed method is verified by sufficient experiments. However, I have the following remarks:
- The developments of research on automatic defect inspection should be summarized in the Introduction.
- The limitations of current research should be analyzed to introduce the proposed method.
- More methods should be selected to verify the performance of the proposed method.
- The performance evaluation metrics of the proposed method are too homogenous and more metrics should be used to discuss its performance.
- The ablution experiments are not sufficient, more relevant experiments should be provided.
- The running time can be discussed to show the superiority of the proposed method.
- Please unify the reference format and font type according to the format requirements of the journal.
Comments on the Quality of English Language
The quality of English language should be improved.
Author Response
Comment 1: The developments of research on automatic defect inspection should be summarized in the Introduction.
Response 1: According to the reviewer’s suggestions, the first two paragraphs (lines 54-91, page 2) have been rewritten in the Introduction for the development on automatic defect inspection using vibration signals.
Comment 2: The limitations of current research should be analyzed to introduce the proposed method.
Response 2: Additionally, several sentences (lines 88-91, page 2) has been added to describe the limitation of current research.
Comment 3: More methods should be selected to verify the performance of the proposed method.
Response 3: Based on our experience and the literature surveyed on defect inspection from the Internet, we have not found any methods for monitoring mold damage using vibration data. Almost all the surveyed papers focus on monitoring wind turbines, motors, electronics, and other applications. There is no benchmark dataset for the damaged mold monitoring. Therefore, we believe it is unfair to compare our proposed method with these studies, as they use different datasets. Additionally, to demonstrate the robustness of our proposed method, we have conducted more ablation experiments, as discussed in Section 3 (lines 390-431, pages 12-13).
Comment 4: The performance evaluation metrics of the proposed method are too homogenous and more metrics should be used to discuss its performance.
Response 4: Based on the reviewer’s comments, several metric criteria have been added to the experiments to assess performance. Additionally, several new paragraphs (lines 360-431, pages 12-13, Section 3) have been added, and Table 1 has been modified to discuss the results of these metrics.
Comment 5: The ablation experiments are not sufficient, more relevant experiments should be provided.
Response 5: According to the reviewer’s suggestion, several ablation experiments have been added as address in Section 3. (lines 390-431, pages 12-13 and the modified Table 1)
Comment 6: The running time can be discussed to show the superiority of the proposed method.
Response 6: To address the reviewer’s concern, we have added a discussion (lines 452-457, page 14, Section 3) on the running time of the proposed method in the revised manuscript. From the discussion, the prediction can be executed in real time on a personal computer.
Comment 7: Please unify the reference format and font type according to the format requirements of the journal.
Response 7: We thank the reviewer for their careful review. The reference format in the revised manuscript has been adjusted according to the requirements of the Sensors journal.
Reviewer 2 Report
Comments and Suggestions for Authors
- The abstract should be restructured to include a short introduction, numerical results and a clear and concise conclusion.
- Page 2, line 48. Figure 1.
- Acronyms must be defined the first time they are used.
- The introduction should be restructured so that it gives a clear idea of the state of the art, further research related to the proposed topic needs to be addressed. On the other hand, the last paragraph should state the main contributions of the paper. Why include Figure 1 in the Introduction? My suggestion would be to include it in the method.
- At the beginning of the methodology it would be desirable to draw a general diagram of the method, add a short paragraph explaining it in general terms and then explain each stage in detail.
- The regression model is explained in very general terms, it needs to be explained in more detail. Why use a regression model?
- Were ablation tests carried out? If yes, authors need to add results
- How do you validate your proposal? Would you be so kind as to explain?
- The discussion needs to be restructured, the results obtained should be compared with those proposed by other authors in the literature.
- What are the limitations of the proposal and what should be addressed in future work?
- It is necessary to include numerical data in the conclusions.
Author Response
Comments 1: The abstract should be restructured to include a short introduction, numerical results and a clear and concise conclusion.
Response 1: The abstract (lines 21-38, page 1) has been rewritten according to the reviewer’s suggestion in this revision.
Comments 2: Acronyms must be defined the first time they are used.
Response 2: We have thoroughly reviewed all acronyms and ensured they are defined upon first use in accordance with the reviewer's comments.
Comments 3: The introduction should be restructured so that it gives a clear idea of the state of the art, further research related to the proposed topic needs to be addressed. On the other hand, the last paragraph should state the main contributions of the paper. Why include Figure 1 in the Introduction? My suggestion would be to include it in the method.
Response 3: The third paragraph (lines 92-100, pages 2-3) in Section 1 has been revised to clearly present the proposed method. Based on our experience, this approach is the first to utilize vibration data for mold health monitoring. Most existing methods focus on rolling bearings and rotating machinery, with limited research on powder metallurgy mold health monitoring. Therefore, the second paragraph in Section 1 (lines 60-91, page 2) has also been revised to better address related research on fault diagnosis across various applications. Additionally, several new paragraph (lines 101-117, page 3) has been added to explicitly outline the contributions of this study. Lastly, Figure 1 has been moved to Section 2 in accordance with the reviewer’s suggestions.
Comments 4: At the beginning of the methodology it would be desirable to draw a general diagram of the method, add a short paragraph explaining it in general terms and then explain each stage in detail.
Response 4:Figure 1 has been moved to Section 2 based on the reviewer’s suggestion. In addition, the first two paragraphs (lines 124-156, pages 3-4) in section 2 and a general diagram (Figure 2) have been added to describe the function of each stage of proposed method.
Comments 5: The regression model is explained in very general terms, it needs to be explained in more detail. Why use a regression model?
Response 5: According to the reviewer’s suggestions, several paragraphs (lines 288-322, pages 9-10) and Figure 8 have been revised to provide a more detailed explanation of the architecture of the proposed model. The input patterns consist of a sequence of temporal patterns. The regression model on temporal patterns is used to analyze sequences of historical patterns for predicting future patterns. If the prediction closely matches the current observation, the mold is determined to be healthy; otherwise, it is classified as abnormal. A new paragraph (lines 276-287, pages 9) has been added to explain why we use a regression model.
Comments 6: Were ablation tests carried out? If yes, authors need to add results
Response 6: We have conducted several ablation experiments, as addressed in Section 3 (lines 390-431, pages 12-13, and the modified Table 1), to demonstrate the effectiveness and robustness of the proposed method.
Comments 7: How do you validate your proposal? Would you be so kind as to explain?
Response 7: To validate our proposal, three toy examples (lines 349-373, pages 11–12) are first illustrated to show the MSE values between the model predictions and the current data during a stamping cycle. Second, ablation experiments (lines 390-431, pages 12–13) were conducted, and the notable results are presented in Table 1. From the observations in Table 1, we find that the errors between the predictions and the current data are small under normal conditions, whereas the errors are large under abnormal conditions. They are clearly separated and the threshold value for issuing an alarm can be easily determined.
Comments 8: The discussion needs to be restructured, the results obtained should be compared with those proposed by other authors in the literature.
Response 8: According to the reviewer’s suggestion, Section 4 (lines 462-493, pages 15) has been rewritten. Our review of the literature found no existing methods for monitoring mold damage using vibration data. Additionally, there is no benchmark dataset available for mold damage monitoring. As a result, we believe it is inappropriate to compare our proposed method with previous studies, as they rely on different datasets. Additionally, to demonstrate the robustness of our proposed method, we have conducted more ablation experiments, as discussed in Section 3.
Comments 9: What are the limitations of the proposal and what should be addressed in future work? It is necessary to include numerical data in the conclusions.
Response 9: The Conclusion (line 495-526, page 16, Section 5) has been rewritten to address the limitation of the proposal and future works. We also include the numerical results in the conclusion.
Reviewer 3 Report
Comments and Suggestions for Authors
Dear Authors,
I have reviewed your work and find the topic of mold damage monitoring using deep learning to be interesting and relevant. However, the manuscript has several weaknesses that need to be addressed before it can be considered for publication.
1. The manuscript's writing quality needs significant improvement. There are numerous grammatical errors, typos, and non clear sentence structures throughout the text. This makes it difficult to understand the authors' ideas and follow the technical details.
2. The manuscript lacks sufficient technical depth in several areas. The description of the BiLSTM model and the attention mechanism is superficial. The authors should provide more details about the model's architecture, hyperparameter settings, and training process.
3. The experimental validation is not comprehensive enough. The authors only use a small dataset collected from a single machine. The results are not compared with other state-of-the-art methods. More experiments are needed to demonstrate the robustness and generalizability of the proposed method.
4. The novelty of the proposed method is not clearly established. The authors should clearly differentiate their work from existing methods and highlight the unique contributions of their approach.
5. The introduction is poorly written and lacks focus. The authors should clearly state the problem, the motivation for their work, and the key contributions of their approach.
6. The related work section is inadequate. The authors only cite a few relevant papers. They should provide a more comprehensive review of the literature on mold damage monitoring and deep learning methods for anomaly detection.
7. The methods section needs more technical details. The authors should provide a more in-depth description of the BiLSTM model and the attention mechanism. They should also explain the rationale behind their design choices.
8. The experimental results section needs to be expanded. The authors should provide more details about the dataset, the experimental setup, and the evaluation metrics. They should also compare their results with other state-of-the-art methods.
9. The discussion section is weak. The authors should discuss the limitations of their approach and potential areas for future work.
Author Response
Comments 1: The manuscript's writing quality needs significant improvement. There are numerous grammatical errors, typos, and unclear sentence structures throughout the text. This makes it difficult to understand the authors' ideas and follow the technical details.
Response 1: The revised manuscript has been edited for clarity by the English Language Editing service of MDPI, according to the reviewer’s comments.
Comments 2: The manuscript lacks sufficient technical depth in several areas. The description of the BiLSTM model and the attention mechanism is superficial. The authors should provide more details about the model's architecture, hyperparameter settings, and training process.
Response 2: According to the reviewer’s suggestions, several paragraphs have been added (lines 288-322, pages 9-10) to describe the architecture of the proposed model.
Comments 3: The experimental validation is not comprehensive enough. The authors only use a small dataset collected from a single machine. The results are not compared with other state-of-the-art methods. More experiments are needed to demonstrate the robustness and generalizability of the proposed method.
Response 3: Since the occurrence of damaged molds is unpredictable and we do not deliberately destroy the molds, it is difficult to collect data from damaged molds. Additionally, to the best of our knowledge and based on the surveyed literature, this study is the first to evaluate mold health using IMU-based vibration data. There is no benchmark dataset available for comparison. Almost all the surveyed studies focus on monitoring wind turbines, motors, electronics, and other applications. We believe it is unfair to compare our proposed method with these studies, as they use different datasets. Therefore, ablation experiments (lines 390-431, page 12-113) have been added to demonstrate the robustness and generalization of the proposed method, in accordance with the reviewer’s suggestion.
Comments 4: The novelty of the proposed method is not clearly established. The authors should clearly differentiate their work from existing methods and highlight the unique contributions of their approach.
Response 4: Similar to the previous comment, to the best of our knowledge, this study is the first one to monitor mold health using vibration data. Most studies focus on fault classification in motors, wind turbines, and other applications. Based on the reviewer’s comments, the paragraph (lines 60-91, page 2) has been rewritten to clearly highlight the differences between our work and existing studies. Additionally, a new paragraph (lines 101-117, page 3) has been added to clearly present our contributions in this study.
Comments 5: The introduction is poorly written and lacks focus. The authors should clearly state the problem, the motivation for their work, and the key contributions of their approach.
Response 5: According the reviewer’s suggestion, we have rewritten the paragraphs in Section 1 (lines 43-122, pages 2-3) to clearly state the problem, the motivation, and the contribution of this study.
Comments 6: The related work section is inadequate. The authors only cite a few relevant papers. They should provide a more comprehensive review of the literature on mold damage monitoring and deep learning methods for anomaly detection.
Response 6: Based on our experience and a review of the literature on defect inspection available online, we have found no methods for monitoring mold damage using vibration data. Nearly all the studies we reviewed focus on monitoring wind turbines, motors, electronics, and other similar applications as described at the second paragraph in Section 1 (lines 60-91, page 2).
Comments 7: The methods section needs more technical details. The authors should provide a more in-depth description of the Bi-LSTM model and the attention mechanism. They should also explain the rationale behind their design choices.
Response 7: The input patterns are composed of a sequence of temporal patterns. A regression model on these temporal patterns is used to analyze historical sequences and predict future patterns. If the predicted pattern closely aligns with the current observation, the mold is considered healthy; otherwise, it is classified as abnormal. In response to the reviewer’s suggestions, we have added several paragraphs (lines 276-322, pages 9-10) to describe the architecture of the proposed model in detail.
Comments 8: The experimental results section needs to be expanded. The authors should provide more details about the dataset, the experimental setup, and the evaluation metrics. They should also compare their results with other state-of-the-art methods.
Response 8: In response to the reviewer’s suggestion, we have conducted additional ablation experiments (lines 390-431, pages 11-12, Section 3) to further demonstrate the robustness of our proposed method. As the responses in Comment 3, there are no existing methods or benchmark datasets for monitoring mold damage using vibration data. Therefore, we believe it is inappropriate to compare our proposed method with previous studies, as they rely on different datasets.
Comments 9: The discussion section is weak. The authors should discuss the limitations of their approach and potential areas for future work.
Response 9: According the reviewer’s suggestion, we have rewritten Sections 4 and 5 (lines 462-516, pages 15-16) to clearly discuss the limitations of this study and the potential areas for future work.
Round 2
Reviewer 1 Report
Comments and Suggestions for Authors
- The author’s reply is very unprofessional. For each question, the pre - and post-revision content should be reflected in the response.
- For problem 3, the proposed method utilized a Bi-LSTM model with an attention mechanism to predict normal stamping vibrations several minutes in advance. Other methods such as GRU, LSTM, DBN, etc. should be selected as comparison methods to predict normal stamping vibration, which can effectively demonstrate the effectiveness of using Bi-LSTM
- For problem 6, only the computation time of the proposed method is discussed, which is not a good indication of the performance of the method. Only by comparing the calculation time of the proposed method with other methods can an objective conclusion be given.
- For question 7, several changes should be listed for reviewers to check, rather than simply stating that it has been changed.
The language expression of this article needs to be improved.
Reviewer 2 Report
Comments and Suggestions for Authors
The comments have been addressed. However, I have two final recommendations: firstly, improve the quality of the images and increase the font size on the graphics to make them easier to read; and secondly, place the images after the paragraph in which they are mentioned.
Author Response
Comments: The comments have been addressed. However, I have two final recommendations: firstly, improve the quality of the images and increase the font size on the graphics to make them easier to read; and secondly, place the images after the paragraph in which they are mentioned.
Response: We sincerely thank the reviewer for their constructive suggestions to improve the quality of our paper. We have enhanced the image quality and increased the font size in all figures to ensure better readability throughout the paper. Figure 3(c) has been redrawn for quality improvement. Additionally, we have verified that all figures are placed immediately after their first mention in the article.
Reviewer 3 Report
Comments and Suggestions for Authors
Dear Author,
Thank you for revising, and re-submitting again. I would like to make the following changes for further improvement of the paper.
- The quality of the figure, and table used in this paper is poor, i would strongly suggest to make the figure in high resolution, and also increase the size of the label
- Still the novelty of the paper is missing, and please try to emphasise the originality
- Proper literature survey is missing, Please add more reference related to the present work
I advise to go for the english proof reading check.
Author Response
Comments 1: The quality of the figure, and table used in this paper is poor, I would strongly suggest to make the figure in high resolution, and also increase the size of the label.
Response 1: We sincerely appreciate the reviewer’s constructive suggestions to enhance the quality of this manuscript. We have improved the image quality to a higher resolution and increased the font size of labels in all figures for better readability. Additionally, Figure 3(c) has been redrawn for quality improvement.
Comments 2: Still the novelty of the paper is missing, and please try to emphasize the originality
Response 2: We thank the reviewer for their suggestions. As mentioned previously, our research lab is the first to utilize vibration data and deep learning methods for mold damage monitoring in the powder metallurgy industry. A new paragraph (lines 155-175, page 4) has been added to clearly present the originality of this study.
Comments 3: Proper literature survey is missing, please add more reference related to the present work
Response 3: We have added several papers based on the reviewer’s suggestions. Additionally, the surveyed deep learning-based methods have been categorized into various network architectures and applications, as listed in Table 1 (page 3). As mentioned in the article, existing deep learning models for anomaly detection in industrial applications primarily focus on rolling bearings and rotating machinery, with limited research on powder metallurgy mold health monitoring (lines 125-128, page 3). Moreover, to the best of our knowledge, no related studies have applied vibration data and deep learning methods to mold damage detection. The second paragraph in Section 1 (lines 60–91, page 2, in the previous version) has been rewritten and expanded into several paragraphs (lines 60–128, pages 2–3, in this revision) to present relevant works on anomaly detection research. The surveyed papers have also been re-ordered in the reference list according to their citation order.
